# Bilateral Pulmonary Embolism in a 12-Year-Old Girl with Steroid-Resistant Nephrotic Syndrome

**DOI:** 10.3390/children7060062

**Published:** 2020-06-15

**Authors:** Osama Y. Safdar, Rahaf H. Rajab, Rand G. Alghanemi, Gazal A. Tantawi, Noora A. Alsulami, Aeshah A. Alsayed, Abdullah K. Habiballah

**Affiliations:** 1Pediatric Nephrology Center of Excellence, Pediatric Department, Faculty of Medicine, King Abdulaziz University, Jeddah 21414, Saudi Arabia; 2Faculty of Medicine, King Abdulaziz University, Jeddah 21414, Saudi Arabia; rahf717@hotmail.com (R.H.R.); Rand.ghz@gmail.com (R.G.A.); Ghazal.tnt@hotmail.com (G.A.T.); Noraalsulami@hotmail.com (N.A.A.); Aisha_Alsayed@hotmail.com (A.A.A.); a.habib0@hotmail.com (A.K.H.)

**Keywords:** nephrotic syndrome, pulmonary embolism, steroid resistance

## Abstract

Nephrotic syndrome is the most common glomerular disease among children. Although most cases respond to steroid therapy, approximately 10–20% of patients exhibit resistance to conventional steroid therapy and are labeled as steroid-resistant. Such patients are at risk of complications, including infection, thrombosis, and chronic kidney disease. Nephrotic syndrome is considered a thrombogenic condition. Pulmonary embolism is associated with high mortality, and early treatment is essential for the survival of patients. Here, we report the case of a 12-year-old girl with late steroid resistance who developed bilateral pulmonary embolism.

## 1. Introduction

Pulmonary embolism is a respiratory condition that is potentially fatal in patients with nephrotic syndrome. Early diagnosis and treatment are essential to improve the prognosis. Steroid-resistant nephrotic syndrome is considered a risk factor for hypercoagulable status owing to non-selective proteinuria and a significant decrease in the expression of anticoagulant proteins, such as protein C and protein S. Thus, patients with steroid-resistant nephrotic syndrome are at a high risk of developing venous thrombosis. There have been a few reported cases of bilateral pulmonary embolism related to nephrotic syndrome in pediatric patients. Here, we report the case of a 12-year-old girl with steroid-resistant nephrotic syndrome who developed bilateral pulmonary embolism. Computed topography (CT) confirmed the diagnosis, and low-molecular-weight heparin (LMWH) was initiated immediately. The patient survived, demonstrating the importance of early diagnosis and initiation of treatment for this fatal condition.

## 2. Case Presentation

A 12-year-old Indonesian girl was diagnosed with nephrotic syndrome at the age of 4 years. From that age, she responded well to oral steroids with occasional relapses. At 12 years of age, she experienced a relapse and did not respond to 4 weeks of oral steroids. Because of the late development of steroid resistance, we performed renal biopsy, which revealed focal segmental glomerulosclerosis. Cyclosporine (50 mg twice per day) was administered, and the patient showed a partial response with a decrease in proteinuria by more than 50%. Three months before the time of writing, she experienced a serious relapse and was started on intravenous methylprednisolone (600 mg/m^2^/day).

She presented to the emergency room of our hospital complaining of shortness of breath, requiring oxygen, and with a decreased level of consciousness. Pulse oximetry at the time of admission revealed tachycardia; however, the patient reported no chest pain. Her medical history was otherwise insignificant with no past history of surgery, blood transfusions, or known allergies. Examination at the time of admission revealed cushioned features (moon face and abdominal striae). The patient weighed 49 kg, and her height was 129 cm. Her vital signs were as follows: temperature, 37.1 °C (axillary); pulse, 120 beats/min(brachial); blood pressure, 124/88 mmHg; and respiratory rate, 24 breaths/min (tachypneic). Cardiovascular examination showed normal heart sounds with no abnormal sounds or murmurs on auscultation, intact pulsation, and good perfusion on palpation. Respiratory examination showed equal bilateral air entry with no abnormal sounds, wheezing, or crepitation. Systemic examination showed no significant factors. The results of laboratory investigations at the time of admission are detailed in Table 1.

CT brain venography obtained 2 days after admission revealed that the cerebral venous system, including both internal jugular veins, both sigmoid and transverse sinuses, the straight sinus, and the inferior and superior sagittal sinuses were well opacified with contrast and no filling defects, ruling out venous sinus thrombosis. Electrocardiography revealed sinus rhythm, and cardiac echocardiography showed very small thread-like structures at the entry of the superior vena cava, which did not cause the obstruction. This may have indicated a normal variant or a very small thrombus for clinical correlation. The CT pulmonary angiogram showed filling defects of the right -middle, lower-lobar, and segmental pulmonary artery and those of the left-upper, lower-lobar, and segmental pulmonary artery. The pulmonary trunk was normal in size, measuring 2.4 cm in diameter. A peripheral truncated consolidation with cystic spaces was noted in the superior segment of the right lower lobe. A thrombus was detected within the inferior vena cava, and there was a right-ventricle-filling defect representing a thrombus. There were no intra- or extra-thoracic lymph nodes of significant size, and no pericardial or pleural effusion was detected (Figure 1). A diagnosis of bilateral pulmonary embolism was made based on these findings.

The patient was started on 0.5 L oxygen until vital and clinical signs were stable, after which enoxaparin (1 mg/kg/dose) was administered subcutaneously, every 12 h. Plasma heparin was monitored by plasma anti-Xa assay, and bilateral lower-limb Doppler ultrasound was used to locate the source of the thrombus. Ultrasound findings were normal.

## 3. Discussion

The pathophysiology of thrombogenesis in nephrotic syndrome is not clearly understood; however, it is known to involve three major aspects. Firstly, it involves genetic background, such as a history of congenital antithrombin deficiency or factor V Leiden (which are known mutations), single-nucleotide polymorphisms associated with the prothrombotic state, or other unknown genetic predispositions. Other than the genetic factors, environmental factors play a major role in thromboembolic events. These factors include medications, inflammation, and the presence of a central venous catheter. Finally, the thrombogenic nature of the disease plays an important part and is mainly due to the leakage of high-molecular-weight proteins, such as albumin, through the primary defect in the glomeruli, which is caused by massive proteinuria [1]. As well as the loss of high-molecular-weight proteins, proteins that regulate coagulation (such as protein S, antithrombin III, plasminogen, and plasmin) are lost, resulting in a state of thrombogenesis [1]. To counteract the changes that occur in the plasma due to protein loss in the urine, the production of hemostatic proteins in the liver increases, resulting in the development of a prothrombotic state. Fibrinogen, which enhances platelet activity and aggregation of red blood cells, increased in patients with nephrotic syndrome. These patients also had higher levels of factor V, factor VII, and alpha-2 macroglobulin, which promote thrombus formation [1].

The patient in the present case report had low serum albumin (5 g/L) levels at the time of admission. Hypoalbuminemia increases thromboxane A2 synthesis, which stimulates platelet adhesiveness [2]. Furthermore, platelet hyperactivity is enhanced by hypercholesterolemia and increased levels of Von Willebrand factor, which also promote platelet adhesion. Hypovolemia increases the viscosity of the blood, resulting in increased aggregation of red blood cells and formation of blood clots [1,3]. Fibrinolysis is also compromised by urinary loss of plasmin, which also contributes to the hypercoagulable state [1]. A combination of these factors is considered to result in the increased incidence of thromboembolic events in patients with nephrotic syndrome [4]. One of the most serious forms of venous thromboembolism is acute pulmonary embolism; however, there are few reports or studies of pulmonary embolism in young adults or children [5].

A cohort study involving 512 patients with nephrotic syndrome (72 of whom were below 18 years old) found that the prevalence of thrombosis is lower among children than among adults with nephrotic syndrome (17% vs. 37%) [6]. The prevalence of thrombosis was also found to be higher among patients with membranous nephropathy [6]. Another study found the prevalence of the condition to be higher among patients with diabetic nephropathy or an unspecified nephrotic syndrome than among patients with other nephrotic syndrome types. This may be due to the different combinations of risk factors for thromboembolism in patients with different types of nephropathies [7]. The patients in this previous study were older with a higher prevalence of hypertension, diabetes, and previous incidence of thromboembolism [7]. Another retrospective study carried out over 22 years involving 447 children with nephrotic syndrome found that 2% experienced thromboembolism; 1.5% had steroid-sensitive nephrotic syndrome, and 3.8% had steroid-resistant nephrotic syndrome, which indicates that steroid-resistant nephrotic syndrome is a risk factor for thromboembolism [8].

Pulmonary embolism is a serious condition in children with nephrotic syndrome. According to the results of previous case reports (Table 2), symptoms can range from asymptomatic to death; therefore, timely treatment and diagnosis are critical for survival. The patient in the present report developed unspecific symptoms of respiratory distress with normal findings on chest radiographs. We administered LMWH to manage pulmonary embolism, which is considered standard pharmacological management for pulmonary embolism.

A meta-analysis of 12 randomized trials (with a total of 1951 patients) compared the administration of fixed-dose subcutaneous LMWH with dose-adjusted intravenous unfractionated heparin as introductory treatments for pulmonary embolism and found the two treatments to be equivalent in terms of safety and efficacy for non-massive pulmonary embolism. The simple administration and lack of obligatory follow-up blood tests make LMWH a very good option for patients with pulmonary embolism. Previous meta-analysis reported that massive bleeding occurred in 1.4% of patients who received LMWH compared with 2.3% who received unfractionated heparin, which represents an insignificant difference.

In many pediatric cases, LMWHs are the anticoagulants, which are selected for initial prophylaxis and the treatment of thromboembolism due to their many advantages. These include an expected pharmacokinetic profile, which decreases the need for monitoring, the possibility of outpatient use due to the long half-life of the drug, the lack of drug–drug or drug–diet interactions, and a decreased risk of complications, such as heparin-induced thrombocytopenia and osteoporosis. The LMWHs that are most frequently used in pediatric patients are enoxaparin, dalteparin, reviparin, and tinzaparin. However, most clinical data on LMWH use in pediatric patients arises from research involving enoxaparin [21]. Enoxaparin has predictable pharmacokinetics and appropriate subcutaneous administration, which is associated with greater bioavailability. It also needs less demanding monitoring than other anticoagulants [29].

In conclusion, bilateral pulmonary embolism is a rare condition which can be fatal when it occurs in children with nephrotic syndrome. Early diagnosis and management are critical to patient survival, and LMWHs are the standard treatment for this potentially fatal condition.

## Figures and Tables

**Figure 1 children-07-00062-f001:**
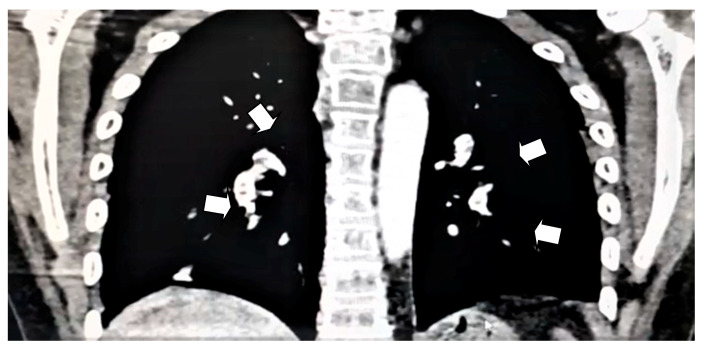
Computed tomography pulmonary angiogram showing filling defects of the right-middle, lower-lobar, and segmental pulmonary artery and of the left-upper, lower-lobar, and segmental pulmonary artery. White arrows indicate the defects.

**Table 1 children-07-00062-t001:** Laboratory results at the time of admission.

Test Name	Result	Unit	Reference Range
White blood cell count	13.86	K/UL	4.5–13.5
Red blood cell count	4.50	M/UL	4–5.40
Hemoglobin	12.6	g/dL	12–15
Hematocrit	37	%	35–49
Mean cell volume	84.0	FL	80–96
Mean cell hemoglobin	28.0	Pg	32–36
Platelet count	328	K/UL	150–450
C-reactive protein	3.13	mg/L	0–3
Prothrombin time	10.8	s	10–13
Activated partial thrombin time	45.4	s	25.1–36.5
D-Dimer	43.783	mg/L	0–0.5
International Normalized Ratio	0.96	Ratio	0.85–1.3
Sodium	137	mmol/L	136–145
Potassium	3.9	mmol/L	3.5–5.1
Chloride	106	mmol/L	98–107
Urea	4.7	mmol/L	2.5–6.4
Creatinine	22	µmol/L	53–115
Total protein	46	g/L	64–82
Albumin	5	g/L	40.2–47.6
Alkaline phosphatase	146	U/L	141–460
Aspartate amino transferase	16	U/L	15–37
Alanine amino transferase	18	U/L	12–78
Gamma glutamyl transferase	155	U/L	5–85
Total bilirubin	2	U/L	0–17
Antinuclear antibody titer	1:640		Negative

**Table 2 children-07-00062-t002:** Summary of previous case reports involving patients who experienced pulmonary embolism.

Age (Years)	Underlining Disease	Presented Symptoms	Prompt Diagnosis of PE	Method of Diagnosis	Prompt Treatment of PE	Outcome	Reference
17	Factor V Leiden nephrotic syndrome	Respiratory tract infection with cough and back pain	Presumptive	Echocardiography	Yes	Improvement and disappearance of pain and cough (lived)	[9]
10	Focal segmental glomerulosclerosis nephrotic syndrome			Autopsy		Delayed death due to saddle pulmonary thromboembolism	[10]
10	Nephrotic syndrome	Shortness of breath	NO	Autopsy	NO	Died	[11]
2.6	Nephrotic syndrome	Pyrexia, vomiting, poor fluid intake, and poorly localized chest/abdominal pain	YES	Pulmonary angiography	YES	Resolved and lived	[12]
12	Nephrotic syndrome	Increasing weight, abdominal pain, and reduced urine output	YES	CTPA with contrast	YES	Resolved and lived	[13]
12	Nephrotic syndrome	Abdominal pain, edema, and diarrhea	YES	CTPA	YES	Resolved and lived	[13]
14	Nephrotic syndrome	Vomiting, watery diarrhea, abdominal pain, and chest pain.	YES	Contrast enhanced CT	YES	Resolved and lived	[14]
6	Nephrotic syndrome	Non-productive cough and dyspnea on exertion	NO	Autopsy	NO	Died	[15]
5	Nephrotic syndrome	Difficulty in breathing and decreased urine output	YES	CTPA	YES	Resolved and lived	[16]
10	Nephrotic syndrome	Sharp chest pain, dyspnea, and perioral cyanosis	NO	Autopsy	NO	Died	[17]
15	Nephrotic syndrome	Signs of complications	NO	Autopsy	NO	Died	[18]
2	Asthmatic bronchitis and nephrotic syndrome	Tachypnea Edema of eyelids and legs Ascites Hypertension	YES	Ventilation–perfusion lung scanning	YES	Resolved and lived	[19]
2	Nephrotic syndrome	Relapse and spontaneous bacterial peritonitis features	YES	Lung perfusion scan	YES	Resolution and lived	[20]
4.5	Nephrotic syndrome	Submandibular swelling, cough, abdominal distention	YES	Lung perfusion scan	YES	PE resolved but she died	[21]
10.5	Nephrotic syndrome	Mild right-sided pleuritic pain	YES	Radionuclide ventilation and perfusion scans	YES	Resolved and lived	[22]
3	Nephrotic Syndrome	Malaise, breathlessness, and tachycardia	YES	Pulmonary angiography	YES	Resolved and lived	[23]
12	Nephrotic Syndrome	Hemoptysis, cough, and shortness of breath	YES	Contrast enhanced CT	YES	Resolved and lived	[24]
3	Nephrotic syndrome	Mild respiratory distress	YES	Ventilation–perfusion lung scanning	YES	Resolved and lived	[25]
10	Nephrotic syndrome	Chest pain and shortness of breath	YES	CTPA	YES	Resolved and lived	[26]
16	Nephrotic syndrome	Severe left-sided chest pain and hemoptysis	YES	CTPA	YES	Resolved and lived	[27]
20	Nephrotic syndrome	Abdominal discomfort and fatigue	YES	CTPA	YES	Resolved and lived	[28]

Abbreviations: CT, computed tomography; CTPA, computed tomography pulmonary angiography; PE, pulmonary embolism.

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
