# Peer review of "Bilateral Pulmonary Embolism in a 12-Year-Old Girl with Steroid-Resistant Nephrotic Syndrome"

_children, 2020, doi:10.3390/children7060062_

Round 1
Reviewer 1 Report
The manuscript is significantly improved.
Reviewer 2 Report
Bilateral Pulmonary Embolism in a 12-Year-Old Female with Steroid-Resistant Nephrotic Syndrome
REVIEWER comments
Reviewer(s)' Comments to Author:
1. The authors described Bilateral Pulmonary Embolism in a 12-Year-Old
Female with Steroid-Resistant Nephrotic Syndrome. I think this manuscript is well-written and comprehensive to the readers.
2. How about the Immune function of this patient?
3. How about “with no past history of surgery other than the renal biopsy, blood
46 transfusions, or known allergies” What is the allergic factor? medicine ?
Author Response
Please see the attachment

This manuscript is a resubmission of an earlier submission. The following is a list of the peer review reports and author responses from that submission.
Round 1
Reviewer 1 Report
Introduction
- Please change “Pulmonary embolismis…” in “Pulmonary embolism..”
Case presentation
- Please change “…occasionally .At 12 years…” in “…occasionally. At 12 years….”
- Please change “…methyprednisoloneat 600 …” in “…methyprednisolone at 600 …”
- Please rewrite a sentence “The systemic review was otherwise…” in “ Her medical hystory…”
- Please change “…sounds ormurmurs…” in “…sounds or murmurs…”
- Please rewrite a sentence removing her…“Laboratory investigations showed that herwhite blood cell count was elevated at 13.86 K/UL, her red blood cell count was 4.50 K/UL, her hemoglobin was 12.6 K/UL, her hematocrit was 37 K/UL, her mean cell volume was 84.0 K/UL, her mean cell hemoglobin was 28.0 FL, her platelet count was 328 K/UL, and her CRP was less than3.13 mg/L.
- How was the value of proteinuria 24 hours?
- The authors report that albumin was 5 g/L. Among laboratory findings,hypoalbuminemia is the most significant inde-pendent predictor factor of thrombotic risk, especially for values <2 g/dL. How the authors could explain that? Please read and add in discussion the reference “Gigante A, Barbano B, Sardo L, Martina P, Gasperini ML, Labbadia R, Liberatori M, Amoroso A, Cianci R. Hypercoagulability and nephrotic syndrome. Curr Vasc Pharmacol. 2014 May;12(3):512-7. Review.
Author Response
Dear editor
Thanks for your valuable comments
Here is my response
All the changes in punctuations done
Regarding low serum albumin and relation to thormbosis , this was addressed and added in discussion
A table with results is done
24 hours urine collection was not done unfortunalty
Reviewer 2 Report
The authors reported a cased of bilateral pulmonary embolism in steroid resistant nephrotic syndrome. However, there are some problems had to be detail clarified
1. We suggested that the introduction was needed to be re-designed again. Authors need to introduce the link of pulmonary embolism and steroid resistant nephrotic syndrome.
- Extensive editing of English language is necessary. The article section had some problem.
- What is the important point or special point of this case? Authors have to report in introduction.
- In case presentation, authors have to mention the initial symptoms, including duration.
- Authors have to design a table of laboratory data with normal range.
- Image studies is necessary for readers. Authors have to provide, including ECG, CT, heart echo, chest Xray…..
- It is necessary to report also all the new studies of the literature on this topic which are now missing.
- The mechanism of pathophysiological links between pulmonary embolism and steroid resistant nephrotic syndrome is reported in some previous studies. We suggested authors to report current results in article.
Author Response
Dear editor
Thanks for your valuable comments
Here are my responses
1-Introductions re designed as per suggestions
2-The article reviewed by native English speaker
3-Done and added in the introduction
4-Done and added to the patient presentation
5-Done and the table is added
6-CT scan added
7-Done and the new studies added in the table
8-Done in the 1st paragraph of the discussion
Round 2
Reviewer 2 Report
We thank author for provided clear information by point to point!